# Training Linear Finite-State Machines

**Arash Ardakani, Amir Ardakani, and Warren J. Gross**
Department of Electrical and Computer Engineering, McGill University, Montreal, Canada
`{arash.ardakani, amir.ardakani}@mail.mcgill.ca`
`warren.gross@mcgill.ca`

## Abstract

A finite-state machine (FSM) is a computation model to process binary strings in sequential circuits. Hence, a single-input linear FSM is conventionally used to implement complex single-input functions , such as tanh and exponentiation functions, in stochastic computing (SC) domain where continuous values are represented by sequences of random bits. In this paper, we introduce a method that can train a multi-layer FSM-based network where FSMs are connected to every FSM in the previous and the next layer. We show that the proposed FSM-based network can synthesize multi-input complex functions such as 2D Gabor filters and can perform non-sequential tasks such as image classifications on stochastic streams with no multiplication since FSMs are implemented by look-up tables only. Inspired by the capability of FSMs in processing binary streams, we then propose an FSM-based model that can process time series data when performing temporal tasks such as character-level language modeling. Unlike long short-term memories (LSTMs) that unroll the network for each input time step and perform back-propagation on the unrolled network, our FSM-based model requires to backpropagate gradients only for the current input time step while it is still capable of learning long-term dependencies. Therefore, our FSM-based model can learn extremely long-term dependencies as it requires $1/l$ memory storage during training compared to LSTMs, where $l$ is the number of time steps. Moreover, our FSM-based model reduces the power consumption of training on a GPU by 33% compared to an LSTM model of the same size.

## 1   Introduction

In the paradigm of deep learning, deep neural networks (DNNs) and recurrent neural networks (RNNs) deliver state-of-the-art accuracy across various non-sequential and temporal tasks, respectively. However, they require considerable amount of storage and computational resources for their efficient deployment on different hardware platforms during both training and inference. In recent years, several techniques have been introduced in literature to address these limitations. To reduce the computational complexity of deep learning models, replacing expensive multiplications with simple operations is a common approach as multiplications dominate neural computations. In XNOR networks [1–6], weights and activations are constrained to only two possible values of $-1$ or 1, replacing multiplications with XNOR operations. Similar to XNOR networks, stochastic computing (SC)-based networks perform neural computations using bitwise operations by representing continuous values as bit streams [6–11]. Even though the aforementioned approaches managed to significantly reduce the complexity of DNNs and RNNs, they fail to completely remove multiplications.

On the other hand, model compression techniques [12–15] and designing compact networks [16–19] are commonly used to reduce the memory requirement of deep learning models. However, even using these techniques cannot solve the memory requirement issue imposed by the nature of RNNs that unroll the network for each time step and store all the intermediate values for backpropagation. For

instance, a long short-term memory (LSTM) [20] of size 1000, which is a popular variant of RNNs, cannot fit into the GeForce GTX 1080 Ti for the step sizes beyond 2000. Moreover, increasing their step sizes severely decreases their convergence rate. Therefore, the current architecture of DNNs and RNNs requires rethinking for their efficient deployment on various hardware platforms.

In this paper, we introduce a method to train finite-state machines (FSMs). An FSM is a mathematical model of computation which is composed of a finite number of states and transitions between those states. The main operation of an FSM involves traversing through a sequence of states in an orderly fashion and performing a predetermined action upon each state transition. Since FSMs are designed to process sequences of data, we use SC, that converts continuous values to bit streams, in order to perform non-sequential tasks using FSMs. Such a network is referred to as FSM-based network. The FSM-based network is composed of weighted linear FSMs (WLFSMs) only and is obtained by stacking multiple layers of them where each WLFSM is connected to every WLFSM in the previous and the next layer. In WLFSMs, each state is associated with a weight and an output is generated by sampling from the weight associated to the present state. The FSM-based network is multiplication-free and is designed to perform the inference computations on bit streams in SC domain only. To train the FSM-based network, we derive a function from the steady state conditions of linear FSMs that computes the occurrence probability for each state of an FSM. We mathematically prove that this function is invertible. The inverse function is then used to derive the derivative of the FSM's computational function w.r.t. its input, allowing to train deep FSM-based networks. we then employ the FSM-based network for two different tasks: synthesis of multi-input complex functions and image classifications. Unlike conventional methods that use a single WLFSM to synthesize single-input complex functions [6, 21, 22], we show that our FSM-based network can approximate multi-input complex functions such as 2D Gabor filters using linear FSMs only. We also adopt our FSM-based network to perform a classification task on the MNIST dataset [23]. We show that our FSM-based network significantly outperforms its SC-based counterparts of the same size in terms of accuracy performance while requiring $1/2$ the number of operations.

We also introduce an FSM-based model to perform temporal tasks. This model is inspired by sequential digital circuits where FSMs are used as a memory (i.e., register) to store the state of the model [24]. In addition to WLFSM, the FSM-based model also consists of fully-connected networks which serve as a transition function and an output decoder similar to combinational logic in sequential circuits. The transition function controls the transition from one state to another whereas the output function performs the decision-making process based on either the present state in *Moore machine*, or both the present state and the present input in *Mealy machine*. Since the next state is determined based on the current input and the present state, gradients are backpropagated for the current time step only. In contrast, two widely-used variations of recurrent neural networks, i.e., LSTMs and gated recurrent units (GRUs) [25], unroll the network for each time step and backpropagate gradients for all the time steps on the unrolled network. As a result, our FSM-based model requires $1/l$ memory elements to store the intermediate values and reduces the power consumption of training on a GPU by 33% compared to an LSTM model of the same size, where $l$ denotes the number of time steps. We show that our FSM-based model can learn extremely long dependencies (e.g., sequence length of 2500) when performing the character-level language modeling (CLLM) task.

## 2 Preliminaries

In SC's numerical system, continuous values are represented as the frequency of ones in random bit streams [26]. In this way, arithmetic computations are performed using simple bit-wise operations on bit streams. Since a single bit-flip in a stochastic stream results in a marginal change in the continuous value represented by the bit stream, SC-based implementations can tolerate small errors. As a result, SC-based systems offer ultra low-cost fault-tolerant hardware implementations for various applications [27]. Given the continuous value $x \in [0, 1]$, its stochastic stream vector $\mathbf{x} \in \{0, 1\}^l$ of length $l$ in SC's *unipolar* format is generated such that

$$\mathbb{E}[\mathbf{x}] = x, \tag{1}$$

where the expected value of the vector $\mathbf{x}$ is denoted by $\mathbb{E}[\mathbf{x}]$. In SC's *bipolar* format, the bipolar value $x^b \in [-1, 1]$ is expressed as

$$\mathbb{E}[\mathbf{x}^b] = (x^b + 1)/2, \tag{2}$$

where $\mathbf{x}^b \in \{0, 1\}^l$. Given two independent stochastic streams of $\mathbf{a}$ and $\mathbf{b}$, multiplications are performed using the bit-wise AND and XNOR operations in unipolar and bipolar formats [27],

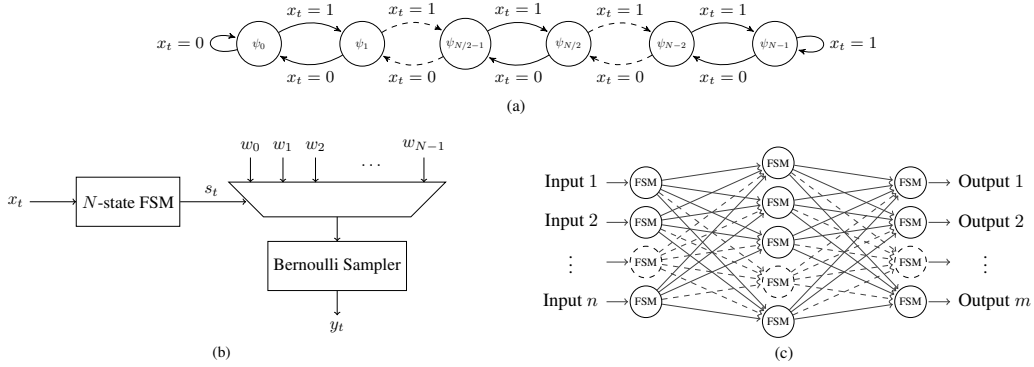

Figure 1: (a) A WLFSM with $N$ states where $x_t$ denotes the $t^{\text{th}}$ entry of the input stream $\mathbf{x} \in \{0,1\}^l$ for $t \in \{1, 2, \ldots, l\}$. (b) An architecture implementing the WLFSM with $N$ states. (c) A general form of an FSM-based network.

respectively. Additions in SC are performed using the scaled adders that fit the result of additions into the permitted intervals of $[0, 1]$ in unipolar format and $[-1, 1]$ in bipolar format. The scaled adder uses a multiplexer to perform an addition between two stochastic streams of $\mathbf{a}$ and $\mathbf{b}$. The multiplexer's output $\mathbf{c}$ is obtained by

$$\mathbf{c} = \mathbf{a} \cdot \mathbf{s} + \mathbf{b} \cdot (1 - \mathbf{s}), \tag{3}$$

where "$\cdot$" denotes the bit-wise AND operation. In this way, the expected value of $\mathbf{c}$ is equal to $(\mathbb{E}[\mathbf{a}] + \mathbb{E}[\mathbf{b}])/2$ when the stochastic stream $\mathbf{s}$ represents the continuous value of 0.5 (i.e., $\mathbb{E}[\mathbf{c}] = 0.5$).

In SC, complex functions are conventionally implemented using linear FSMs [28]. A linear FSM consists of a finite of states arranged in a linear form. A general form of a linear FSM with a set of $N$ states (i.e., $\{\psi_0, \psi_1, \ldots, \psi_{N-1}\}$) is illustrated in Figure 1(a). In fact, a linear FSM can be viewed as a saturating counter that cannot increment or decrement beyond its maximum or minimum state value, respectively. The state transitions in linear FSMs are occurred according to the current entry of the input stream. If the current entry (i.e., $x_t \in \{0, 1\}$ for $t \in \{0, 1, \ldots, l\}$) of the input stream $\mathbf{x} \in \{0, 1\}^l$ is 0, the state value is decremented; it is incremented otherwise.

As the first attempt to implement complex functions, Brown and Card in [28] introduced two FSM-based functions: tanh and exponentiation functions. They showed that if an $N$-state linear FSM outputs 1 for the state values greater than or equal to $N/2$ and outputs 0 for other cases, the expected value of the FSM's output approximates $\tanh(Nx/2)$ for the input stochastic stream of $\mathbf{x}$ representing the continuous value of $x$. Similarly, $\exp(-2Gx)$ can be approximated when the linear FSM outputs 1 for the state value less than $N - G$; it generates 0 otherwise. Li *et al.* introduced WLFSMs whose states are associated with a weight [22]. In WLFSMs, a binary output is generated by sampling from the weight associated to the current state as shown in Figure 1(b). To implement a given single-input function using a WLFSM, Li *et al.* formulated the computations as a quadratic programming problem and used numerical methods to obtain the weights. Ardakani *et al.* recently exploited linear regression to obtain the FSM's weights [6]. It was shown that using the regression-based method outperforms the numerical synthesis method in terms of the mean-squared error (MSE). It is worth mentioning that the aforementioned methods were used to synthesize a single WLFSM implementing a single-input complex function.

## 3   FSM-Based Networks

In this section, we introduce a method that allows to backpropagate gradients in FSM-based networks. The FSM-based network is organized into layers of WLFSMs where each WLFSM is connected to every WLFSM in the previous and the next layer with an exception of the first layer. In the first layer of FSM-based networks, each input is solely connected to a single WLFSM as illustrated in Figure 1(c). The inputs of each WLFSM unit are first added and scaled to fit into the permitted SC's range using the scaled adder. The output of the scaled adder is then passed to a WLFSM. In this way, the core computations of FSM-based networks are additions and weight indexing operations. The weight

indexing operations of WLFSMs are implemented using look-up tables (LUTs), bringing a significant benefit for hardware implementations on LUT-based devices such as GPUs and field-programmable gate arrays (FPGAs).

## 3.1 Backpropagation Method

To backpropagate gradients through FSM-based networks, let us first formulate the forward computations of a WLFSM with $N$ states as

$$y_t = \text{Bernoulli}\left(\frac{w_{s_t} + 1}{2}\right), \tag{4}$$

where $s_t \in \{0, 1, \ldots, N-1\}$ and $y_t \in \{0, 1\}$ are the state value and the output value that correspond to the input $x_t \in \{0, 1\}$ at time $t$ for $t \in \{1, 2, \ldots, l\}$. The FSM is also associated with a set of weights (i.e., $\{w_0, w_1, \ldots, w_{N-1}\}$). Performing the computations for every single input entry of the input stochastic vector $\mathbf{x} \in \{0, 1\}^l$ yields a stochastic output vector $\mathbf{y} \in \{0, 1\}^l$ representing the continuous value $y \in \mathbb{R}$ in bipolar format such that $y = 2 \times \mathbb{E}(\mathbf{y}) - 1$. However, training FSM-based networks on stochastic streams is $l\times$ slower than the conventional full-precision training methods. Therefore, we train FSM-based networks on the continuous values of stochastic streams while the inference computations are still performed on stochastic bit streams.

Given the occurrence probability (i.e., the selection frequency) of the state $\psi_i$ as $p_{\psi_i}$ for $i \in \{0, 1, \ldots, N-1\}$, we can also obtain the continuous value of WLFSM's output (i.e., $y \in [-1, 1]$) by

$$y = \sum_{i=0}^{N-1} p_{\psi_i} \times w_{\psi_i}, \tag{5}$$

when the length of stochastic streams goes to infinity (i.e., $l \to \infty$). In the steady state, the probability of the state transition from $\psi_{i-1}$ to $\psi_i$ must be equal to the probability of the state transition from $\psi_i$ to $\psi_{i-1}$, that is

$$p_{\psi_i} \times (1 - p_x) = p_{\psi_{i-1}} \times p_x, \tag{6}$$

where $p_x$ is $(x+1)/2$. In other words, the weight associated to $\psi_i$ is selected with the probability of $p_x$ during the forward state transition (i.e., the state transition from $\psi_{i-1}$ to $\psi_i$) whereas the weight associated to $\psi_{i-1}$ is selected with the probability of $1 - p_x$ during the backward state transition (i.e., the state transition from $\psi_i$ to $\psi_{i-1}$). Therefore, the derivative of the probability of the forward transition w.r.t. $x$ is 1 while the derivative of the probability of the backward transition w.r.t. $x$ is $-1$ in the steady state. In other words, we have

$$\frac{\partial p_{\psi_i}}{\partial x} = -\frac{\partial p_{\psi_{i-1}}}{\partial x}. \tag{7}$$

Moreover, the occurrence probability of all the states must sum up to unity, i.e.,

$$\sum_{i=0}^{N-1} p_{\psi_i} = 1. \tag{8}$$

Based on Eq. (6) and Eq. (8), the general form of the occurrence probability is expressed by

$$p_{\psi_i} = \frac{\left(\frac{p_x}{1 - p_x}\right)^i}{\sum_{j=0}^{N-1} \left(\frac{p_x}{1 - p_x}\right)^j}. \tag{9}$$

Given Eq. (5) and Eq. (9), we can learn the WLFSM's weights to implement a single-input complex function using linear regression. To employ WLFSMs in a multi-layer network, we also need to find the derivative of $p_{\psi_i}$ w.r.t. the input $x$. To this end, we first compute the inverse function for $p_{\psi_i}$, which is then used to obtain the derivative of $p_{\psi_i}$ w.r.t. the continuous value of the input stream $\mathbf{x}$. To find the inverse function for $p_{\psi_i}$, we trained a single WLFSM to implement a linear function whose outputs are its inputs. We observed that the weights of the WLFSM with $N$ states are alternately $-1$ and 1, that is

$$x = \sum_{i=0}^{N-1} (-1)^{i+1} p_{\psi_i}, \tag{10}$$

for $i \in \{0, 1, \ldots, N-1\}$. To prove the validity of Eq. (10), we use the geometric series [29] defined as follows:

$$\sum_{i=0}^{N-1} r^i = \frac{1 - r^N}{1 - r},\tag{11}$$

$$\sum_{i=0}^{N-1} (-1)^{i+1} r^i = \frac{(-1)^N r^N - 1}{1 + r},\tag{12}$$

where $r$ is the common ratio. Using Eq. (11) and Eq. (12), we can rewrite the right side of Eq. (10) as

$$\sum_{i=0}^{N-1} (-1)^{i+1} p_{\psi_i} = \sum_{i=0}^{N-1} \frac{(-1)^{i+1} \left(\frac{p_x}{1-p_x}\right)^i}{\sum_{j=0}^{N-1} \left(\frac{p_x}{1-p_x}\right)^j} = \frac{1}{\sum_{j=0}^{N-1} \left(\frac{p_x}{1-p_x}\right)^j} \sum_{i=0}^{N-1} (-1)^{i+1} \left(\frac{p_x}{1-p_x}\right)^i$$

$$= \frac{1 - \frac{p_x}{1-p_x}}{1 - \left(\frac{p_x}{1-p_x}\right)^N} \times \frac{(-1)^N \left(\frac{p_x}{1-p_x}\right)^N - 1}{1 + \frac{p_x}{1-p_x}}$$

$$= \frac{1 - (-1)^N \left(\frac{p_x}{1-p_x}\right)^N}{1 - \left(\frac{p_x}{1-p_x}\right)^N} \times (2p_x - 1) \stackrel{\text{for even } N}{=} (2p_x - 1) = x.\tag{13}$$

So far, we have provided a mathematical proof for a hypothesis (i.e., Eq. (10)) obtained by synthesizing a linear function using a WLFSM. By taking a derivative w.r.t. $x$ from Eq. (8) and Eq. (10), we obtain the following set of differential equations:

$$\frac{\partial p_{\psi_i}}{\partial x} = -\frac{\partial p_{\psi_{i-1}}}{\partial x}, \quad \sum_{i=0}^{N-1} \frac{\partial p_{\psi_i}}{\partial x} = 0, \quad \sum_{i=0}^{N-1} (-1)^{i+1} \frac{\partial p_{\psi_i}}{\partial x} = 1,\tag{14}$$

for $i \in \{0, 1, \ldots, N-1\}$. By solving the above set of equations, we obtain the derivative of $p_{\psi_i}$ w.r.t. $x$ as

$$\frac{\partial p_{\psi_i}}{\partial x} = \frac{(-1)^{i+1}}{N},\tag{15}$$

which is used to backpropagate gradients in FSM-based networks. The details of training algorithm are provided in Appendix A.

## 3.2   Applications of FSM-Based Networks

As the first application of FSM-based networks, we synthesize 2D Gabor filters. During training, we perform the forward computations using Eq. (9) while the backward computations are performed using Eq. (15). On the other hand, the inference computations of FSM-based networks are performed on stochastic streams. The imaginary part of a 2D Gabor filter is defined as

$$g_{\sigma,\gamma,\theta,\omega}(x, y) = \exp\left(-\frac{\overline{x}^2 + \gamma^2 \overline{y}^2}{2\sigma^2}\right) \sin(2\omega\overline{x}),\tag{16}$$

where $\overline{x} = x\cos\theta + y\sin\theta$ and $\overline{y} = -x\sin\theta + y\cos\theta$. The parameters $\sigma$, $\gamma$, $\theta$ and $\omega$ respectively denote the standard deviation of the Gaussian envelope, the spatial aspect ratio, the orientation of the normal to the parallel stripes of the Gabor filter and the spatial angular frequency of the sinusoidal factor. Figure 2(a-f) shows the simulation results of FSM-based networks implementing a set of 2D Gabor filters used in HMAX model [30]. To obtain the simulation results of Figure 2(a-f), we trained a three-layer FSM-based network of size 4 (i.e., the network configuration of $2 - 4 - 4 - 1$) where each WLFSM contains four states (i.e., $N = 4$). Such a network contains 10 4-state WLFSMs and 112 weights. We used the MSE as our loss function and Adam as the optimizer with the learning rate of 0.1. We also used the total of $2^{20}$ points evenly distributed among inputs for our simulations. It

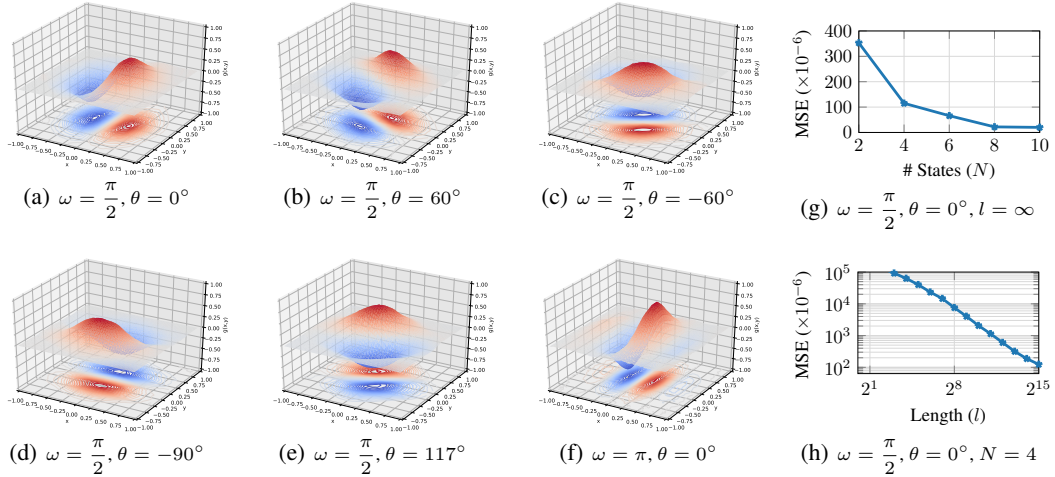

Figure 2: (a-f) The simulation results of 2D Gabor filters with different configurations for $\sigma^2 = 0.125$ and $\gamma = 1$. (g-h) The effect of the stream length and the number of states on the performance of the 2D Gabor filters with the parameters of $\sigma^2 = 0.125$, $\gamma = 1$, $\omega = \dfrac{\pi}{2}$ and $\theta = 0°$.

is worth mentioning that our three-layer FSM-based network implementing the 2D Gabor filters in Figure 2(a-f) approximately yields the MSE of $1 \times 10^{-4}$ when performing the computations with $l = 2^{15}$. To show the effect of stream length and the number of states, we examined our three-layer FSM-based network implementing the 2D Gabor with the parameters of $\sigma^2 = 0.125$, $\gamma = 1$, $\omega = \pi/2$ and $\theta = 0°$ as shown in Figure 2(g-h). The simulation results show that the MSE decreases as the stream length and the number of state increase. Please see Appendix B for more details on model architectures and training settings. In [31], an SC-based implementation of 2D Gabor filters was introduced by approximating the sinusoidal part in Eq. (16) with several tanh functions. In this way, the sine function can be implemented using linear FSMs as discussed in Section 2. It was shown that such an approach requires a 256-state FSM for the exponential part and 9 56-state FSMs for the sinusoidal part with the stream length of $2^{18}$ to obtain a similar MSE to ours. However, this approach is limited to the functions that can be approximated by either tanh or exponentiation function only, whereas our FSM-based network is a general solution to implement any arbitrary target function.

As the second application of FSM-based networks, we perform an image classification task on the MNIST dataset. The details of model architectures and training settings for the classification task are provided in Appendix B. Table 1 summarizes the misclassification rate of two FSM-based networks with different configurations when performing the inference computations on stochastic streams of length 128 (i.e., $l = 128$). According to Table 1, our FSM-based networks significantly outperform the existing SC-based counterparts in terms of the misclassification rate and the required stream length. Moreover, our FSM-based networks require $1/2$ the number of operations than the conventional SC-based implementations of the same size. The choice of using two states and the stream length of 128 for our FSM-based networks was made based on Figure 3 that illustrates their misclassification rate for different number of states and stream lengths. As expected by the nature of stochastic computing, the misclassification error decreases as the stream length increases. For stream lengths greater than 64, the error rate roughly stays the same, making the stream length of 128 as the sweet spot. The simulation results also show that the two-state FSM-based networks perform better than the ones with larger number of states. This observation suggests that using a smaller number of states better regularizes the parameters of the network for this particular task.

## 4 An FSM-Based Model for Temporal Tasks

A state machine consists of three essential elements: a transition function, an output decoder and a memory unit [24]. The memory unit is used to store the state of the machine. The output decoder generates a sequence of outputs based on the present state in Moore machines. The transition function

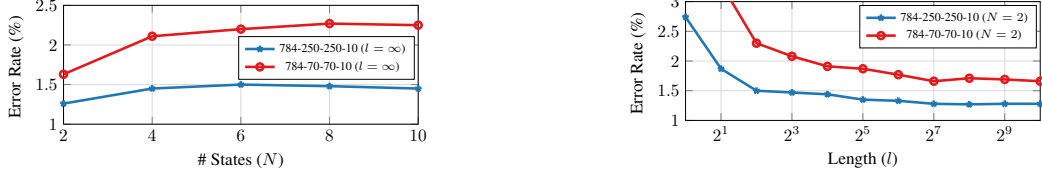

Figure 3: The effect of using different number of states and different stream lengths on the misclassification rate of FSM-based networks on the test set.

determines the next state of the machine based on the present state and the present input. In digital systems, the memory unit is implemented using registers whereas both the transition function and the output decoder are realized using combinational logic. Inspired by the state machine used in sequential circuits, we introduce an FSM-based model that is capable of processing sequences of data. In our FSM-based model, we implement the transition function and the output decoder using a single fully-connected layer. We also use an FSM-based layer as the memory unit. In fact, fully-connected layers and WLFSMs can be viewed as combinational logic and registers, respectively.

## 4.1 Feed-Forward Computations

In our FSM-based model, we use a Moore machine that performs the decision-making process based on the present state only. An $N$-state FSM-based model performs its feed-forward as follows:

$$\mathbf{z} = \mathbf{x}_t \mathbf{W}_x + \mathbf{b}_x, \tag{17}$$

$$\mathbf{s}_t = \text{Clamp}\left(\mathbf{s}_{t-1} + 2 \times \text{Bernoulli}\left(\frac{\mathbf{z}+1}{2}\right) - 1, 0, N-1\right), \tag{18}$$

$$\mathbf{o} = \text{One\_Hot\_Encoder}(\mathbf{s}_t), \tag{19}$$

$$\mathbf{q} = \text{Sigmoid}(\alpha \mathbf{o} \mathbf{W}_o + \mathbf{b}_o), \tag{20}$$

$$\mathbf{y} = \mathbf{q} \mathbf{W}_y + \mathbf{b}_y, \tag{21}$$

where $\mathbf{W}_x \in \mathbb{R}^{d_x \times d_h}$, $\mathbf{W}_o \in \mathbb{R}^{Nd_h \times d_h}$ and $\mathbf{W}_y \in \mathbb{R}^{d_h \times d_y}$ are the weights. The parameters of $\mathbf{b}_x \in \mathbb{R}^{d_h}$, $\mathbf{b}_o \in \mathbb{R}^{d_h}$ and $\mathbf{b}_y \in \mathbb{R}^{d_y}$ denote the biases. The input, containing temporal features at time $t$, is denoted by $\mathbf{x}_t$. The vectors $\mathbf{z} \in \mathbb{R}^{d_h}$ and $\mathbf{y} \in \mathbb{R}^{d_y}$ are the output of the transition function and the output decoder, respectively. The vector $\mathbf{q} \in \mathbb{R}^{d_h}$ is the FSM-based layer's output. Each entry of the vector $\mathbf{s}_t \in \{0, 1, \ldots, N-1\}^{d_h}$ holds the state value of each WLFSM in the FSM-based layer at the time step $t$. The Clamp function replaces the values greater than $N-1$ and the values less than 0 with $N-1$ and 0, respectively. The One\_Hot\_Encoder function converts each entry of the vector $\mathbf{s}_t$ to a one-hot encoded vector of size $N$ and concatenates the one-hot encoded vectors to form the sparse vector of $\mathbf{o} \in \mathbb{R}^{Nd_h}$ such that $\sum_{i=j \times N}^{(j+1) \times N} o_i = 1$, where $o_i$ denotes the $i^{th}$ entry of the vector $\mathbf{o}$ for $j \in \{0, 1, \ldots, d_h - 1\}$. The parameter $\alpha$ is a fixed coefficient that prevents the values of the weight matrix $\mathbf{W}_o$ to become very small. We set $\alpha$ to $d_h^{-1}$ for our simulations in this paper.

Table 1: Performance of our FSM-based network compared to SC-based implementations on the test set of the MNIST dataset.

| Model | Configuration | $(N, l)$ | (# Op., # Weights) | Error Rate (%) |
|---|---|---|---|---|
| FSM-based Network | 784-250-250-10 | (2, 128) | (0.52M, 0.52M) | 1.28 |
| FSM-based Network | 784-70-70-10 | (2, 128) | (0.12M, 0.12M) | 1.66 |
| TCAD'18 [10] | 784-128-128-10 | (NA, $\infty$) | (0.24M, 0.12M) | 3 |
| TCOMP'18 [11] | 784-200-100-10 | (NA, 256) | (0.36M, 0.18M) | 2.05 |
| TVLSI'17 [8] | 784-300-600-10 | (NA, 256) | (0.84M, 0.42M) | 2.01 |

## 4.2 Backpropagation

The challenging step during the training process of the FSM-based model is to derive the derivative of the vector $\mathbf{o}$ w.r.t. $\mathbf{z}$. The gradient of the rest of computations (i.e., the matrix-vector multiplications in Eq. (17), Eq. (20) and Eq. (21)) can be easily obtained using the chain rule. In the FSM-based layer, using one-hot encoded vectors (i.e., Eq. (19)) is to ensure that only the weights associated to the present state of WLFSMs are selected. Each selected weight is a result of either the forward transition (i.e., the transition from $\psi_{i-1}$ to $\psi_i$) when the Bernoulli function outputs 1 or the backward transition (i.e., the transition from $\psi_i$ to $\psi_{i-1}$) when the Bernoulli function outputs -1. It is worth mentioning that $\psi_i$ denotes the $i^{th}$ state of a WLFSM with $N$ states for $i \in \{0, 1, \ldots, N-1\}$. Therefore, the weights associated to the state $\psi_i$ are selected with the probability of $p_z$ during the forward transition and with the probability of $1 - p_z$ during the backward transition, where $p_z$ is the probability of the WLFSM's input $z \in [-1, 1]$ to be 1 (i.e., $p_z = (1+z)/2$). Given the aforementioned information, we have

$$\frac{\partial \mathbf{s}_t}{\partial \mathbf{z}} = \begin{cases} 1 & \text{when Bernoulli} \left( \dfrac{\mathbf{z}+1}{2} \right) == 1 \\ -1 & \text{otherwise} \end{cases}. \tag{22}$$

During the backpropagation through the One_Hot_Encoder function, only the gradients associated to the present state of WLFSMs are backpropagated, that is

$$\hat{s}_{t_j} = \sum_{i=j \times N}^{(j+1) \times N} (o_i \times \hat{o}_i), \tag{23}$$

where $\hat{s}_{t_j}$ is the $j^{th}$ entry of the gradient vector $\hat{\mathbf{s}}_t \in \mathbb{R}^{d_h}$ at the input of the One_Hot_Encoder function for $j \in \{0, 1, \ldots, d_h - 1\}$ and $\hat{o}_i$ the $i^{th}$ entry of the gradient vector $\hat{\mathbf{o}} \in \mathbb{R}^{Nd_h}$ at the output of the One_Hot_Encoder function for $j \in \{0, 1, \ldots, Nd_h - 1\}$. The details of the training algorithm are provided in Appendix C.

## 4.3 Simulation Results

As discussed earlier, states of our FSM-based model are updated based on the present input only. More precisely, the transition function either increments or decrements the state of each WLFSM based on the input features at time $t$. In fact, FSM-based model can be viewed as a time-homogeneous process where the probability of transitions are independent of $t$. As a result, given a temporal task that makes a decision at each time step (e.g., the CLLM task), the backpropagation of the FSM-based model is performed at the end of each time step. In this way, the storage required to store the intermediate values during the training stage is significantly reduced by a factor of $l\times$, allowing to process extremely long data sequences using the FSM-based model. This is in striking contrast to LSTMs where their network is unrolled for each time step and the backpropagation is applied to the unrolled network. The sequence length of LSTMs is thus limited to a few hundreds during the training process since the storage required to store the intermediate values of the unrolled network can easily go beyond the available memory of today's GPUs. For instance, Figure 4 shows the memory usage of the LSTM model versus the FSM-based model and their corresponding test accuracy performance on the GeForce GTX 1080 Ti for different numbers of time steps when both models have the same number of weights and use the batch size of 100 for the CLLM on the Penn Treebank dataset [32]. The simulation results show that the memory usage of our FSM-based model is independent of the number of time steps, making our model suitable for on-chip learning on mobile devices with limited storage. On the other hand, despite a slight performance improvement of the LSTM model, it cannot fit into the GPU for the step sizes beyond 2000. In addition to the less memory requirement of our FSM-based models, they are less computationally intensive as the backward process of each time step in our FSM-based models obviously requires less number of computations than that of all the time steps in the LSTM models. The immediate outcome of the less memory and computation requirements is less power consumption of the GPU. More precisely, training FSM-based models of size 1000 with batch size of 100 roughly draws 160W for all the time steps ranging from 100 to 2500, whereas training the LSTM models of the same size consumes power ranging from 205W to 245W based on our measurements obtained from the NVIDIA system management interface. As the final point, increasing the number of times steps severely impacts the convergence rate of the LSTM model whereas the convergence rate of the FSM-based model remains unchanged (see Figure 4).

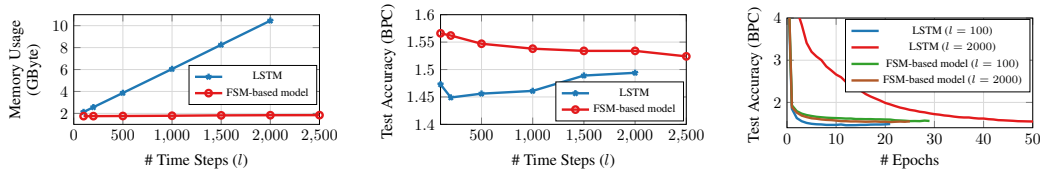

Figure 4: The memory usage and the test accuracy performance of an LSTM model with 1000 hidden states versus a 4-state FSM-based mode of size 1000 (i.e., $d_h = 1000$) for different numbers of time steps and training epochs when performing the CLLM task on the Penn Treebank corpus.

To demonstrate the capability of our FSM-based model in processing temporal data, we perform the CLLM task on Penn Treebank [32], War & Peace [33] and Linux kernel [33] corpora where the performance is measured in terms of bit per character (BPC). The simulation results of our FSM-based model are summarized in Table 2. According to the experimental results, our FSM-based model with 4-state FSMs achieves a comparable accuracy in terms of BPC compared to the LSTM model when both models have the same number of parameters. It is worth mentioning that we set the number of hidden nodes of all the models to 1000 (i.e., $d_h = 1000$) for the Penn Treebank corpus and 500 (i.e., $d_h = 500$) for the War & Peace and the Linux Kernel corpora to obtain the simulation results reported in Table 2. Moreover, our FSM-based network requires $1/7$ the number of operations compared to the LSTM model of the same size as the computational core of the WLFSM's layer involves indexing and accumulate operations only. The model architectures and training settings for all the simulations of the CLLM task are detailed in Appendix D.

## 5    Conclusion

In this paper, we introduced a method to train WLFSMs. WLFSMs are computation models that can process sequences of data. To perform non-sequential tasks using WLFSMs, we used SC that coverts continuous values to bit streams. The networks containing WLFSMs only and performing their computations on stochastic bit streams are call FSM-based networks. As the first application of FSM-based networks, we implemented 2D Gabor filters using FSMs only. As the second application of FSM-based networks, we performed a classification task on the MNIST dataset and we showed that our FSM-based networks significantly outperforms their conventional SC-based implementations in terms of both the misclassification error and the number of operations. As the final contribution of this paper, we introduced an FSM-based model that can perform temporal tasks. We showed that the required storage for training our FSM-based models is independent of the time steps as opposed to LSTMs. As a result, our FSM-based model can learn extremely long data dependencies while reducing the required storage for the intermediate values of training by a factor of $l\times$, the power consumption of training by 33% and the number of operations of inference by a factor $7\times$.

## Broader Impact

Training deep learning models is both financially and environmentally an expensive process. From the environmental point view, it is estimated that the carbon footprint required for developing and training a single deep learning model (e.g., stacked LSTMs) can create 284 tonnes of carbon dioxide, which is equivalent to the lifetime $CO_2$ emissions of five average cars [34]. The environmental impact of training deep learning models is calculated by the total power required to train each model multiplied by the training time spent for its development. On the other hand, the financial impact of training deep learning models is associated to the cost of hardware (e.g., cloud-based platforms, GPUs) and electricity. As a result, using computationally efficient algorithms as well as energy efficient hardware

Table 2: Performance of our FSM-based model when performing the CLLM task on the test set.

| Model | Penn Treebank | | | War & Peace | | | Linux Kernel | | |
|---|---|---|---|---|---|---|---|---|---|
| | # Weights | # Op. | BPC | # Weights | # Op. | BPC | # Weights | # Op. | BPC |
| 4-State FSM-based Model | 4.1M | 1.1M | 1.52 | 1.1M | 0.3M | 1.89 | 1.1M | 0.3M | 1.93 |
| LSTM (Our implementation) | 4.1M | 8.1M | 1.45 | 1.1M | 2.1M | 1.83 | 1.1M | 2.1M | 1.85 |

is of paramount importance to reduce both environmental and financial impacts of deep learning [34]. The first part of this paper focuses on complexity reduction of the inference computations where FSM-based networks are presented. FSM-based networks use SC to perform the inference computations on bit streams. As discussed in Section 2, SC offers ultra-low power implementations that can significantly reduce the cost of specialized hardware for the inference computations. As a result, our FSM-based networks have positive environmental and financial impacts by saving electricity and $CO_2$ emissions. In the last part of the paper, we introduced FSM-based models that can learn extremely long data dependencies while significantly reducing the number of operations and the storage required for training. It is worth mentioning that power consumption of deep learning models are dominated by their memory accesses to the main memory of hardware platforms such as GPUs. To show the impact of our FSM-based model during training, we measured its power consumption when performing the CLLM task on the Penn Treebank corpus. An FSM-based model of size 1000 ($d_h = 1000$) draws 160 W for the step size of 2000 on the GeForce GTX 1080 Ti whereas an LSTM model of the same size requires 245 W on average. Moreover, our FSM-based model converges faster than the LSTM model by at least a factor of $2\times$ (see Figure 4), significantly reducing the training time. Given the power and the training time reductions that our FSM-based model offers, $CO_2$ emissions are reduced by at least a factor of $3\times$ for the given example. Therefore, our FSM-based models contribute remarkably in reduction of carbon dioxide emissions and have positive impacts on the climate change.

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
