[Supplementary Material]

## Appendix A

In this section, we detail the training and the inference computations of FSM-based networks. In FSM-based networks, the training computations are performed using single-precision floating-point format whereas the inference computations are performed using stochastic bit streams. The forward computations of training are detailed in Algorithm 1. In the forward computations of training, we compute the occurrence probability of each state using Eq. (9). Although the occurrence probability of each state can also be obtained by performing the forward computations of training on stochastic bit streams, this approach is time consuming and makes the training procedure very slow. To make sure that the value of weights lies within the bipolar interval of SC during inference, we restrict weights to get values between $-1$ and $1$ during the forward propagation of training. The backward computations of training are detailed in Algorithm 2. In the backward computations of training, the gradients of FSM-based layers are backpropagated using Eq. (10). The loss function $\mathbb{C}$ is chosen based on the target task in FSM-based networks. It is worth mentioning that the number of states (i.e., $N$) in FSM-based networks can take even natural numbers only (see Eq. (13)).

---

**Algorithm 1:** Pseudo code of the forward computations of training in FSM-based networks. $L$ is the number of layers including the output layer. The training loss is denoted as $\mathbb{C}$. $N$ denotes the number of states in FSMs. The Clamp function replaces the values greater than 1 and less than $-1$ with 1 and $-1$, respectively.

**Data:** An input minibatch of $\mathbf{x}^0 \in [-1,+1]^{d_b \times d_{x^0}}$, a target minibatch of $\overline{\mathbf{y}} \in [-1,+1]^{d_b \times d_{x^L}}$, the occurrence probability of the state $\psi_i$ as $\mathbf{p}_{\psi_i}^k \in [0,1]^{d_b \times d_{x^k}}$, the occurrence probability of all the state as $\mathbf{p}_\psi^k \in [0,1]^{d_b \times N d_{x^k}}$ and weights $\mathbf{W}^k \in [-1,+1]^{d_{x^k} \times d_{x^{k+1}}}$ for $k \in \{0, \ldots, L-1\}$ and $i \in \{0, \ldots, N-1\}$.

1 **for** $k = 0 : L - 1$ **do**
2     **for** $i = 0 : N - 1$ **do**
3        $\mathbf{p}_{\psi_i}^k = \dfrac{\left(\dfrac{1+\mathbf{x}^k}{1-\mathbf{x}^k}\right)^i}{\sum_{j=0}^{N-1}\left(\dfrac{1+\mathbf{x}^k}{1-\mathbf{x}^k}\right)^j}$
4     **end**
5     $\mathbf{p}_\psi^k = [\mathbf{p}_{\psi_0}^k, \mathbf{p}_{\psi_1}^k, \ldots, \mathbf{p}_{\psi_{N-1}}^k]$
6     $\mathbf{x}^{k+1} = \dfrac{\left(\mathbf{p}_\psi^k \text{Clamp}(\mathbf{W}^k, -1, 1)\right)}{d_{x^k}}$
7 **end**
8 Compute loss $\mathbb{C}$ given $\mathbf{x}^L$ and $\overline{\mathbf{y}}$

---

**Algorithm 2:** Pseudo code of the backward computations of training in FSM-based networks. $L$ is the number of layers including output layer. The training loss is denoted as $\mathbb{C}$. $N$ and $\eta$ denote the number of states in FSMs and the learning rate, respectively. The gradient of parameters w.r.t. $\mathbb{C}$ is denoted by "ˆ" over their corresponding symbols.

**Data:** Gradients of activations as $\hat{\mathbf{x}}^k \in \mathbb{R}^{d_b \times d_{x^k}}$, the occurrence probability of the state $\psi_i$ as $\hat{\mathbf{p}}_{\psi_i}^k \in \mathbb{R}^{d_b \times d_{x^k}}$, the occurrence probability of all the state as $\hat{\mathbf{p}}_\psi^k \in \mathbb{R}^{d_b \times N d_{x^k}}$ and weights as $\hat{\mathbf{W}}^k \in \mathbb{R}^{d_{x^k} \times d_{x^{k+1}}}$ for $k \in \{0, \ldots, L-1\}$ and $i \in \{0, \ldots, N-1\}$.

1 Compute $\hat{\mathbf{x}}^L = \dfrac{\partial \mathbb{C}}{\partial \mathbf{x}^L}$ given $\mathbf{x}^L$ and $\overline{\mathbf{y}}$
2 $\hat{\mathbf{W}}^{L-1} = \mathbf{x}^{L-1^T}\hat{\mathbf{x}}^L$
3 **for** $k = L - 1 : 1$ **do**
4     $\hat{\mathbf{p}}_\psi^k = \dfrac{1}{d_{x^k}}\hat{\mathbf{x}}^{k+1}\mathbf{W}^{k^T}$
5     $\hat{\mathbf{x}}^k = \sum_{i=0}^{N-1}\dfrac{(-1)^{i+1}}{N}\hat{\mathbf{p}}_{\psi_i}^k$
6     $\hat{\mathbf{W}}^{k-1} = \mathbf{x}^{k-1^T}\hat{\mathbf{x}}^k$
7 **end**
8 **for** $k = 0 : L - 1$ **do**
9     $\mathbf{W}^k \leftarrow \text{Update}(\mathbf{W}^k, \hat{\mathbf{W}}^k, \eta)$
10 **end**

---

Unlike the training computations, the inference computations are performed on stochastic bit streams. Algorithm 3 details the inference computations of FSM-based networks. Since the FSMs' output vector $\mathbf{o}_t^k$ is one-hot encoded, its multiplication with a binary sample of weights $\mathbf{W}^k$ is in fact indexing operations. More precisely, the main computations of inference involve indexing and add operations. Therefore, FSM-based networks are multiplication-free. Moreover, no separate nonlinear activation function is required when using FSM-based networks. In fact, FSMs can be viewed as nonlinear activation functions that can approximate their required non-linearity during training. It is worth mentioning that FSMs are conventionally used to approximate nonlinear functions such as tanh and exponentiation functions in SC domain (see Section 2). Given the aforementioned points, FSM-based

networks are well-suited for applications requiring ultra-low cost implementations of the inference computations.

---

**Algorithm 3:** Pseudo code of the inference computations of FSM-based networks. $L$ is the number of layers including the output layer whereas $l$ denotes the length of stochastic streams. $N$ denotes the number of states in FSMs. The Clamp function replaces the values greater than $N-1$ and less than $0$ with $N-1$ and $0$, respectively. The One_Hot_Encoder function converts each entry of the vector $\mathbf{s}_t$ to a one-hot encoded vector of size $N$ and concatenates the one-hot encoded vectors to form the sparse vector of $\mathbf{o}^k \in \{0,1\}^{d_b \times N d_{x^k}}$ such that $\sum_{i=j \times N}^{(j+1) \times N} o_i^k = 1$, where $o_i^k$ denotes the $i^{th}$ entry of the second dimension of the vector $\mathbf{o}^k$ for $j \in \{0, 1, \ldots, d_{x^k} - 1\}$.

---

**Data:** An input minibatch of $\mathbf{x}^0 \in [-1, +1]^{d_b \times d_{x^0}}$, an output minibatch of $\mathbf{y} \in [-1, +1]^{d_b \times d_{x^L}}$, the state vector of $\mathbf{s}_t^k \in \{0, \ldots, N-1\}^{d_b \times d_{x^k}}$, FSMs' output of $\mathbf{o}_t^k \in \{0,1\}^{d_b \times N d_{x^k}}$, activations of $\mathbf{x}_t^{k+1} \in \{0,1\}^{d_b \times d_{x^k}}$ and weights of $\mathbf{W}^k \in [-1, +1]^{d_{x^k} \times d_{x^{k+1}}}$ for $k \in \{0, \ldots, L-1\}$ and $i \in \{0, \ldots, N-1\}$.

1 $\mathbf{s}_0 = \dfrac{N}{2}$

2 $\mathbf{y} = 0$

3 **for** $t = 1 : l$ **do**

4      $\mathbf{x}_t^0 = \text{Bernoulli}\left(\dfrac{\mathbf{x}^0 + 1}{2}\right)$

5      **for** $k = 0 : L - 1$ **do**

6          $\mathbf{s}_t^k = \text{Clamp}\left(\mathbf{s}_{t-1}^k + 2 \times \mathbf{x}_t^k - 1, 0, N - 1\right)$

7          $\mathbf{o}_t^k = \text{One\_Hot\_Encoder}(\mathbf{s}_t^k)$

8          $\mathbf{x}_t^{k+1} = \text{Bernoulli}\left(\dfrac{\mathbf{o}_t^k \text{Bernoulli}\left(\dfrac{\mathbf{W}^k + 1}{2}\right)}{d_{x^k}}\right)$

9      **end**

10      $\mathbf{y} = \mathbf{y} + \dfrac{2 \times \mathbf{x}_t^L - 1}{l}$

11 **end**

---

# Appendix B

In this section, we provide more details on model architectures and training settings used to synthesize 2D Gabor filters and perform image classification in Section 3.2.

**Synthesis of 2D Gabor Filters**: To implement 2D Gabor filters, we used a three-layer FSM-based network of size 4 where each hidden layer contains 4 WLFSMs (i.e., the network configuration of $2 - 4 - 4 - 1$). In our FSM-based network, we adopted 4-state WLFSMs. Therefore, the total of 10 4-state WLFSMs with 112 weights are required to form our FSM-based network. Since the main goal of this task is to approximate 2D Gabor filters, we used the MSE as our loss function $\mathbb{C}$. To train our FSM-based network, we used the total of $2^{20}$ points evenly distributed in the 2D plane of inputs (i.e., $x$ and $y$ in Eq. (16)). We also used Adam as the optimizer, the learning rate (LR) of $0.1$ and the batch size (BS) of $2^{10}$ during training. Once the FSM-based network was trained, we performed the inference computations on the same points that were used for training to obtain the simulation results

Table 3: The training and the inference settings used in our simulations of 2D Gabor filters.

| Simulation | FSM-Based Network | | Training Parameters | | | | | Inference Parameters |
| --- | --- | --- | --- | --- | --- | --- | --- | --- |
| | Configuration | # States ($N$) | Loss ($\mathbb{C}$) | Optimizer | LR ($\eta$) | BS | # Epochs | Stream Length ($l$) |
| Figure 2(a-f) | $2 - 4 - 4 - 1$ | 4 | MSE | Adam | 0.1 | $2^{10}$ | 1000 | $2^{15}$ |
| Figure 2(g) | $2 - 4 - 4 - 1$ | 2,4,8,10 | MSE | Adam | 0.1 | $2^{10}$ | 1000 | $\infty$ |
| Figure 2(h) | $2 - 4 - 4 - 1$ | 4 | MSE | Adam | 0.1 | $2^{10}$ | 1000 | $2^1, 2^2, \ldots, 2^{15}$ |

Table 4: The training and the inference settings used in our simulations to perform the image classification task on the MNIST dataset.

| Simulation | FSM-Based Network | | Training Parameters | | | | | | Inference Parameters |
|---|---|---|---|---|---|---|---|---|---|
| | Configuration | # States ($N$) | Loss ($\mathbb{C}$) | Optimizer | LR ($\eta$) | BS | # Epochs | Dropout | Stream Length ($l$) |
| Table 1 | 784-250-250-10 | 2 | Cross-Entropy | Adam | 0.05 | 100 | 500 | 0.15 | 128 |
| Table 1 | 784-70-70-10 | 2 | Cross-Entropy | Adam | 0.05 | 100 | 500 | 0.15 | 128 |
| Figure 3 (left subfigure) | 784-250-250-10 | $2, 4, 6, 8, 10$ | Cross-Entropy | Adam | 0.05 | 100 | 500 | 0.15 | $\infty$ |
| Figure 3 (left subfigure) | 784-70-70-10 | $2, 4, 6, 8, 10$ | Cross-Entropy | Adam | 0.05 | 100 | 500 | 0.15 | $\infty$ |
| Figure 3 (right subfigure) | 784-250-250-10 | 2 | Cross-Entropy | Adam | 0.05 | 100 | 500 | 0.15 | $2^1, 2^2, \ldots, 2^{10}$ |
| Figure 3 (right subfigure) | 784-70-70-10 | 2 | Cross-Entropy | Adam | 0.05 | 100 | 500 | 0.15 | $2^1, 2^2, \ldots, 2^{10}$ |

in Figure 2. It is worth mentioning that we used the stream length of $2^{15}$ (i.e., $l = 2^{15}$) to perform the inference computations of our FSM-based network in Figure 2(a-f). Table 3 summarizes the training and the inference settings used in our simulations.

**Image Classification**: As the second application of FSM-based networks, we performed an image classification task on the MNIST dataset. The MNIST dataset of handwritten digits contains 60,000 gray-scale $28 \times 28$ images as a training set and 10,000 as a test set. For our simulations, we used the last 10,000 images of the training set as a validation set. To obtain the simulation results reported in Table 1 and Figure 3, we adopted two three-layer FSM-based networks of size 250 and 70 (i.e., the network configurations of $784 - 70 - 70 - 10$ and $784 - 250 - 250 - 10$). We trained our FSM-based networks using Adam optimizer, the batch size of 100 and the learning rate of 0.1. We also applied the dropout rate of 0.15 to the hidden layers (i.e., dropping 15% of hidden layers' nodes) of our FSM-based networks during training. Since the main goal of this task is to predict a label for a given image, we use the cross-entropy loss function (i.e., the cross-entropy function on top of the softmax output as the loss function). It is worth mentioning that we reported the test error rates as the results of our FSM-based networks in Table 1 and Figure 3. The detailed training and inference settings of our FSM-based networks for the image classification task on the MNIST dataset is provided in Table 4.

# Appendix C

In this section, we detail the training method of our FSM-based models. Before training the FSM-based model for the given number of time steps (i.e., $l$), we initialize the state values of FSMs with $\lfloor N/2 \rfloor$. Unlike the FSM-based networks, the number of states (i.e., $N$) in FSM-based models can take any natural number. The transition function and the output decoder perform fully-connected computations according to Eq. (17) and Eq. (21), respectively. In the memory unit (i.e., the FSM-based layer), the state values are either incremented or decremented by stochastically sampling from inputs to this layer according to Eq. (18). In the training procedure of FSM-based models, the Bernoulli function with a fixed seed must be used during both the forward propagation (i.e., Eq. (18)) and the backward propagation (i.e., Eq. (22)) of training. The reason of using a fixed seed is to obtain the same transition direction of the forward propagation during the backward propagation of training. Algorithm 4 provides the detailed training method of our FSM-based models. As discussed in Section 4, gradients are backpropagated and parameters are updated at the end of each time step in FSM-based models (see Algorithm 4). It is worth mentioning that the state values can also be updated deterministically. More precisely, the Sign function can be used instead of the Bernoulli function in the deterministic approach. Both the stochastic and the deterministic approaches result in a same accuracy performance. However, using the Sign function results in a faster training as it is less computationally intensive than the Bernoulli function.

# Appendix D

In this section, we provide details on training settings and model architectures used to obtain the results reported in Table 2 and Figure 4. We performed the CLLM task using our FSM-based model on three different corpora: Penn Treebank (PT), War & Peace (WP), and Linux Kernel (LK).

**Penn Treebank**: We split the Penn Treebank corpus into 5017k, 393k and 442k training, validation and test sets, respectively. The Penn Treebank corpus has the character size of 50. For this task, we used an FSM-based model of size 1000 (i.e., $d_h = 1000$). The cross entropy loss was minimized on minibatches of size 100 while using ADAM learning rule. We used a learning rate of 0.05.

**Algorithm 4:** Pseudo code of the training algorithm for FSM-based models. $l$ is the number of time steps. The training loss is denoted as $\mathbb{C}$. $N$ and $\eta$ denote the number of states in FSMs and the learning rate, respectively. The gradient of parameters w.r.t. $\mathbb{C}$ is denoted by " $\hat{}$ " over their corresponding symbols. The Clamp function replaces the values greater than $N - 1/1$ and less than $0/-1$ with $N - 1/1$ and $0/-1$, respectively. $\hat{\sigma}$ denotes the derivative of the Sigmoid function. The One_Hot_Encoder function converts each entry of the vector $\mathbf{s}_t$ to a one-hot encoded vector of size $N$ and concatenates the one-hot encoded vectors to form the sparse vector of $\mathbf{o} \in \{0,1\}^{d_b \times N d_h}$ such that $\sum_{i=j \times N}^{(j+1) \times N} o_i = 1$, where $o_i$ denotes the $i^{th}$ entry of the second dimension of the vector $\mathbf{o}$ for $j \in \{0, 1, \ldots, d_h - 1\}$. The parameter $\alpha$ is set to $d_h^{-1}$. " $\times$ " denotes element-wise multiplications. Note that $d_x$ is equal to $d_y$ in the CLLM task.

---

**Data:** An input minibatch of $\mathbf{X} \in \mathbb{N}^{d_b \times d_x \times l}$, an input minibatch of $\mathbf{x}_t \in \mathbb{N}^{d_b \times d_x}$ at the time step $t$, a target minibatch of $\overline{\mathbf{Y}} \in \mathbb{N}^{d_b \times d_x \times l}$, a target minibatch of $\overline{\mathbf{y}}_t \in \mathbb{N}^{d_b \times d_x}$ at the time step $t$, the transition function's output $\mathbf{z} \in [-1, 1]^{d_b \times d_h}$, the transition function's weights $\mathbf{W}_x \in \mathbb{R}^{d_x \times d_h}$, the transition function's biases $\mathbf{b}_x \in \mathbb{R}^{d_h}$, the state vector of $\mathbf{s}_t \in \{0, \ldots, N-1\}^{d_b \times d_h}$, the FSMs' output of $\mathbf{o} \in \{0,1\}^{d_b \times N d_h}$, the FSM-based layer's output of $\mathbf{q} \in \mathbb{R}^{d_b \times d_h}$, the FSM-based layer's weights $\mathbf{W}_o \in \mathbb{R}^{N d_h \times d_h}$, the FSM-based layer's biases $\mathbf{b}_o \in \mathbb{R}^{d_h}$, the output decoder's output $\mathbf{y} \in \mathbb{R}^{d_b \times d_y}$, the output decoder's weights $\mathbf{W}_y \in \mathbb{R}^{d_h \times d_y}$, the output decoder's biases $\mathbf{b}_y \in \mathbb{R}^{d_y}$, the gradient of the transition function's output $\hat{\mathbf{z}} \in \mathbb{R}^{d_b \times d_h}$, the gradient of the transition function's weights $\hat{\mathbf{W}}_x \in \mathbb{R}^{d_x \times d_h}$, the gradient of the transition function's biases $\hat{\mathbf{b}}_x \in \mathbb{R}^{d_h}$, the gradient of the state vector $\hat{\mathbf{s}}_t \in \{0, \ldots, N-1\}^{d_b \times d_h}$, the gradient of the FSMs' output $\hat{\mathbf{o}} \in \{0,1\}^{d_b \times N d_h}$, the gradient of the FSM-based layer's output $\hat{\mathbf{q}} \in \mathbb{R}^{d_b \times d_h}$, the gradient of the FSM-based layer's weights $\hat{\mathbf{W}}_o \in \mathbb{R}^{N d_h \times d_h}$, the gradient of the FSM-based layer's biases $\hat{\mathbf{b}}_o \in \mathbb{R}^{d_h}$, the gradient of the output decoder's output $\hat{\mathbf{y}} \in \mathbb{R}^{d_b \times d_y}$, the gradient of the output decoder's weights $\hat{\mathbf{W}}_y \in \mathbb{R}^{d_h \times d_y}$ and the gradient of the output decoder's biases $\hat{\mathbf{b}}_y \in \mathbb{R}^{d_y}$ for $t \in \{1, \ldots, l\}$.

---

1   $\mathbf{s}_0 = \lfloor \dfrac{N}{2} \rfloor$

2   **for** $t = 1 : l$ **do**

3     $\mathbf{x}_t = \mathbf{X}[:, :, t]$

4     $\overline{\mathbf{y}} = \overline{\mathbf{Y}}[:, :, t]$

5     $\mathbf{z} = \text{Clamp}\,(\mathbf{x}_t \mathbf{W}_x + \mathbf{b}_x, -1, 1)$

6     $\mathbf{s}_t = \text{Clamp}\left(\mathbf{s}_{t-1} + 2 \times \text{Bernoulli}\left(\dfrac{\mathbf{z}+1}{2}\right) - 1, 0, N-1\right)$

7     $\mathbf{o} = \text{One\_Hot\_Encoder}(\mathbf{s}_t)$

8     $\mathbf{q} = \text{Sigmoid}(\alpha \mathbf{o} \mathbf{W}_o + \mathbf{b}_o)$

9     $\mathbf{y} = \mathbf{q} \mathbf{W}_y + \mathbf{b}_y$

10    $\mathbf{h} = \text{Softmax}(\mathbf{y})$

11    $\mathbb{C} = \text{Cross\_Entropy}(\mathbf{h}, \overline{\mathbf{y}}_t)$

12    $\hat{\mathbf{y}} = \dfrac{\partial \mathbb{C}}{\partial \mathbf{y}} = \overline{\mathbf{y}}_t - \mathbf{h}$

13    $\hat{\mathbf{q}} = \hat{\mathbf{y}} \mathbf{W}_y^T$

14    $\hat{\mathbf{W}}_y = \mathbf{q}^T \hat{\mathbf{y}}$

15    $\hat{\mathbf{o}} = \alpha(\hat{\sigma}(\alpha \mathbf{o} \mathbf{W}_o + \mathbf{b}_o) \times \hat{\mathbf{q}}) \mathbf{W}_o^T$

16    $\hat{\mathbf{W}}_o = \alpha \mathbf{o}^T (\hat{\sigma}(\alpha \mathbf{o} \mathbf{W}_o + \mathbf{b}_o) \times \hat{\mathbf{q}})$

17    $\hat{s}_{t_j} = \sum_{i=j \times N}^{(j+1) \times N} (o_i \times \hat{o}_i)$

18    $\hat{\mathbf{z}} = \hat{\mathbf{s}}_t \times \left(2 \times \text{Bernoulli}\left(\dfrac{\mathbf{z}+1}{2}\right) - 1\right)$

19    $\hat{\mathbf{W}}_x = \mathbf{x}_t^T \hat{\mathbf{z}}$

20    $\mathbf{W}_y \leftarrow \text{Update}(\mathbf{W}_y, \hat{\mathbf{W}}_y, \eta)$

21    $\mathbf{b}_y \leftarrow \text{Update}(\mathbf{b}_y, \hat{\mathbf{y}}, \eta)$

22    $\mathbf{W}_o \leftarrow \text{Update}(\mathbf{W}_o, \hat{\mathbf{W}}_o, \eta)$

23    $\mathbf{b}_o \leftarrow \text{Update}(\mathbf{b}_o, \hat{\sigma}(\alpha \mathbf{o} \mathbf{W}_o + \mathbf{b}_o) \times \hat{\mathbf{q}}, \eta)$

24    $\mathbf{W}_x \leftarrow \text{Update}(\mathbf{W}_x, \hat{\mathbf{W}}_x, \eta)$

25    $\mathbf{b}_x \leftarrow \text{Update}(\mathbf{b}_x, \hat{\mathbf{z}}, \eta)$

26   **end**

Table 5: The training settings used in our simulations to perform the CLLM task on the Penn Treebank, War & Peace and Linux Kernel datasets.

| Simulation/Dataset | Network Configuration | | | Training Parameters | | | | | | |
|---|---|---|---|---|---|---|---|---|---|---|
| | Model | # States ($N$) | Size ($d_h$) | Loss ($\mathbb{C}$) | Optimizer | LR ($\eta$) | BS | # Epochs | Dropout | Time Step ($l$) |
| Figure 4 (left subfigure)/PT | FSM-based | 4 | 1000 | Cross-Entropy | Adam | 0.05 | 100 | 500 | 0.15 | $100 - 2500$ |
| Figure 4 (left subfigure)/PT | LSTM | NA | 1000 | Cross-Entropy | Adam | 0.001 | 100 | 500 | 0 | $100 - 2000$ |
| Figure 4 (middle subfigure)/PT | FSM-based | 4 | 1000 | Cross-Entropy | Adam | 0.05 | 100 | 500 | 0.15 | $100 - 2500$ |
| Figure 4 (middle subfigure)/PT | LSTM | NA | 1000 | Cross-Entropy | Adam | 0.001 | 100 | 500 | 0 | $100 - 2000$ |
| Figure 4 (right subfigure)/PT | FSM-based | 4 | 1000 | Cross-Entropy | Adam | 0.05 | 100 | 29 | 0.15 | 100 |
| Figure 4 (right subfigure)/PT | FSM-based | 4 | 1000 | Cross-Entropy | Adam | 0.05 | 100 | 25 | 0.15 | 2000 |
| Figure 4 (right subfigure)/PT | LSTM | NA | 1000 | Cross-Entropy | Adam | 0.001 | 100 | 21 | 0 | 100 |
| Figure 4 (right subfigure)/PT | LSTM | NA | 1000 | Cross-Entropy | Adam | 0.001 | 100 | 50 | 0 | 2000 |
| Table 2/PT | FSM-based | 4 | 1000 | Cross-Entropy | Adam | 0.05 | 100 | 500 | 0.15 | 2500 |
| Table 2/PT | LSTM | NA | 1000 | Cross-Entropy | Adam | 0.001 | 100 | 500 | 0 | 100 |
| Table 2/WP | FSM-based | 4 | 500 | Cross-Entropy | Adam | 0.05 | 100 | 500 | 0.15 | 2000 |
| Table 2/WP | LSTM | NA | 500 | Cross-Entropy | Adam | 0.001 | 100 | 500 | 0 | 100 |
| Table 2/LK | FSM-based | 4 | 500 | Cross-Entropy | Adam | 0.05 | 100 | 500 | 0.15 | 2000 |
| Table 2/LK | LSTM | NA | 500 | Cross-Entropy | Adam | 0.001 | 100 | 500 | 0 | 100 |

**Linux Kernel and War & Peace**: Linux Kernel and Leo Tolstoy's War & Peace corpora consist of 6,206,996 and 3,258,246 characters and have the character size of 101 and 87, respectively. We split Linux Kernel corpus into 4566k, 621k and 621k and War & Peace corpus into 2932k, 163k and 163k training, validation and test sets, respectively. We used an FSM-based model of size 500. ADAM learning rule was used as the update rule with the learning rate of 0.05. The cross entropy loss was minimized on minibatches of size 100 for the CLLM task on these two datasets.

To train our FSM-based models, we followed the training method detailed in Algorithm 4. Table 5 shows the training settings used to report the results in Table 2 and Figure 4. It is worth mentioning that we applied the dropout rate of $0.15$ to the last layer (i.e., dropping 15% of output decoder's nodes) of our FSM-based networks during training. Moreover, we used the number of time steps (i.e., the sequence length of $l$) that results in the best BPC for both the LSTM and the FSM-based models in Table 2.