[Reviews · NeurIPS 2020]

Review 1

Summary and Contributions: This paper presents NN architectures in which the basic units of information and performed operations are not scalar values and linear translations/ nonlinear activations but rather binary streams (representing real numbers in the range [0,1] as the fraction of times their bit is 'on') and "weighted linear finite state machines" (WLFSMs), which convert between binary streams. The conversion between values using these machines is more expressive than simple linear transformations, for instance, they may also compute tanh or exponents. The authors compute the derivative of the outputs of WLFSMs with respect to their inputs, enabling connecting them into a multi-layer architecture through which backpropagation can be applied. The authors describe how to connect these WLFSMs into architectures for both image processing (which they evaluate on MNIST) and temporal tasks (evaluating on the Penn Treebank). They show that the amount of memory needed for training these architectures is much lower, making training much lighter, and enabling longer-range dependencies.

Strengths: The paper derives backpropagation for WLFSMs, enabling connecting them into novel 'NN' architectures in which the base computation is not linear transformations but rather the (different) range of computing power enabled by WLFSMs. I am very curious whether this will find a different range of expressive power, possibly enabling computation of functions previously out of our range. Additionally, the paper claims much lighter backpropagation for a sequence-processing architecture that they have built for these models, this would be very beneficial if it turns out to also be practical (i.e., train well in practice).

Weaknesses: The exact construction of the architectures from the WLFSM units is hard to understand, and I am unclear on how image or sequence data is expressed and passed into the presented models (and the appendix does not clarify this). Hopefully this can be fixed with clarifications from the authors, as the results do seem exciting if the claims about number of computations/memory needed for training are correct. A sketch of each of the architectures and how they apply to image and temporal data (respectively) would help immensely. I elaborate more in the 'clarity' section. In general, the paper is rather difficult for me, though admittedly I have no experience with WLFSMs or stochastic computing.

Correctness: The derivation of the backpropagation seems correct and the trick using the inverse and multiple qualities of the derivatives/weights was even quite cool, though I feel unequipped to verify it perfectly. I have not understood the exact way in which the model processes images or sequences and so cannot tell if the construction is 'reasonable'.

Clarity: The paper took some passes to understand, though this may be because I am not familiar with stochastic computing. Specific notes: 1. It would especially help to more clearly describe the architectures used for image processing and for sequence processing, especially the latter. An image showing the inputs, outputs (and their 'types') and the connection between the different WLFSMs would be very helpful. In particular, I am still unclear on how exactly the input for a sequence is encoded: suppose I am trying to pass in the character-level sequence 'abc'. What is passed into the network? An example, even a toy architecture, would be really helpful. 2. Additionally it is not entirely clear how bit streams are pooled (lines 124-125 seem to suggest that the stream may take on values other than 1 or 0 at each step, which seems confusing to me?. As a concrete question: if one WLFSM 'A' is receiving input from 3 other WLFSMs, and at some time step t two of the input WLFSMs have transmitted '1' and the other has transmitted '0', what is passed into 'A'? (is it '1', '0', '2/3', or a randomly (uniformly) sampled bit from one of the streams, eg '1' with probability 2/3 and '0' with probability 1/3)? In particular if two streams are being passed into another WLFSM and one of them is transmitting '1' 100% of the time, will that 'drown out' the information in the other? 3. For the temporal tasks, it seems very odd to have a model whose backpropagation/update is based on the current time step only. How can a 'meaningful' state transition (i.e. one in which the state learns to encode useful information from past occurences) be encouraged in this setup? 4. This paper seems to constantly be converting between values in the range (0,1) and (-1,1), 'littering' the equations with remnants of these conversions ((x+1)/2 instead of x, etc). Is it possible to somehow normalise all discussed values at the beginning, so that the remainder of the paper only deals with values in one range? It would make it much easier to follow. 5. My understanding is that WLFSMs essentially compute functions from the range (0,1) to the range (0,1). It does seem that these functions are more expressive than linear transformations (eg lines 105-109), but it would help to explain why it is valuable to use the binary-stream representation for these fractions as opposed to directly passing them into and out of the WLFSMs? (Maybe this is what happens in the implementation in practice?)

Relation to Prior Work: I am not familiar enough with WLFSMs and surrounding literature to judge the relationship of this work to others. With respect to temporal data, it would probably be better to compare to a standard implementation of an LSTM (eg, that provided by the pytorch library). For the comparison to MNIST, it might also have been worthwhile to note a comparison to other NN-based implementations in addition to SC-based ones. These comments are not critical, just suggestions.

Reproducibility: No

Additional Feedback: For the sequence-processing model, you frequently note that the update/computation are based only on current state, making your model much lighter than RNN (which has to backpropagate through multiple time steps). Does this mean the state-update of the model is never learned, i.e., the state transitions of this model are fully deterministic, and it is only the outputs of each state are being learned? If so, is there not a risk that a single state may be reached multiple times by different prefixes, on which it has to learn very different behaviour? Otherwise -- i.e., if the state transition is being learned -- then how is the model learning this without backpropagating through time? I would really appreciate clarifications on this. ==== I have read the author response and appreciate the clarifications. For presentation I wonder if it may still be cleaner to discuss everything in 0/1 or -1/1 exclusively, and note the parts of the architecture that would undergo conversion in practice.


Review 2

Summary and Contributions: This paper presents a method that can train a multi-layer FSM-based network. Furthermore, they developed an FSM-based model to handle time series data. Compared with LSTM, this method only requires 1/L memory storage and effectively reduces the power consumption when training on GPU.

Strengths: 1) Compared with LSTM, this model dose a better work both in memory usage and test accuracy. 2) This model can directly update via backpropagation 3) Sufficient theoretical elaboration and analysis.

Weaknesses: In addition to LSTM, have you compared your method with others?

Correctness: yes

Clarity: yes

Relation to Prior Work: yes

Reproducibility: Yes

Additional Feedback: UPDATE: After reading the rebuttal, only part of concerns have been resolved. So, I still keep my point.


Review 3

Summary and Contributions: - Introduce a method to train multi-layer Finite State Machine networks. - Prove the invertibility of a derived function (from FSM steady state conditions), allowing them to compute derivatives. - Demonstrate that these models can: - Synthesize multi-input complex functions (2D Gabor filters) - Outperform SC counterparts of the same size with only half the operations required. - Perform Image classification tasks that require holding a representation of partial input ‘in memory’ - Process and perform well on time-series and sequential data: Character Level Language Modeling on Penn Treebank, War and Peace, and Linux Corpus - Demonstrate computational pros of these models: - No multiplication, only require look-up tables at inference time for the simpler models. - For time series, only requires back propagating gradients for the current input time step, making memory requirements O(1) in sequence length. - Reduces power consumption vs. LSTMs of the same ‘size’ by 33% - Reduces number of operations required for inference by 7x

Strengths: Proofs of validity for gradient computations. Empirical evaluations demonstrate improvement over other stochastic-computing counterpart models. Empirical evaluations demonstrate such models can achieve good performance on temporal-MNIST Empirical evaluations demonstrate similar performance to similarly sized single-layer LSTMs on character level language modeling across several corpora. Both empirical evaluations and math demonstrate significant memory gains for long sequences.

Weaknesses: The number of training examples for learning a single gabor filter seems large: 2^20 ~= 1 million. It would strengthen the result to see performance for a smaller set of training examples. The comparison to LSTMs could be made stronger by investigating multi-layer LSTMs with the same number of parameters. These may have better performance on the CLLM task. The authors refer to the model's ability to use long-term dependencies by pointing to performance on temporal-mnist and character level language modeling. In order to more specifically investigate the ability of these models to use long-term dependencies, I would suggest evaluation on permuted MNIST, as a comparison to temporal mnist where the pixels are in order. Additionally I would suggest evaluations on synthetic tasks meant to test ability to use long-term dependencies (e.g. a copying task).

Correctness: Proofs seem to be correct. Empirical evaluations span both simple function approximation as well as a temporal image classification task and language modeling. Empirical evaluations could be improved by evaluating on synthetic tasks and on permuted MNIST.

Clarity: Yes. Paper does a good job of explaining models, derivations, and experimental setup.

Relation to Prior Work: Mostly. Related work could be improved by citing and referring to work on truncated backpropagation through time (both heuristically set, and adaptively set - see example here: https://arxiv.org/pdf/1905.07473.pdf), as well as work on contractive recurrent backpropagation (see an example here: http://proceedings.mlr.press/v80/liao18c/liao18c.pdf) .

Reproducibility: Yes

Additional Feedback: L41-42 should be rephrased. Lots of work has investigated efficient deployment of DNNs, either by compression, training binary networks, or working with lower-bit representations (e.g. float8). L75-76: I think you mean a memory cost of 1/l in comparison to LSTM? Figure 1: Can you explain if dashed connections in (c) have meaning? L139: Does using stochastic bit streams at inference time affect inference time? It would be great to see benchmarks of time taken for inference. L291-222: Do you have an intuition for why FSM epochs-to-convergence is affected less by longer inputs than for an LSTM? ----- I've read the author response. I appreciate the extra information and clarifications.

[Author Response · NeurIPS 2020]

We thank the reviewers for their careful reading of our manuscript and their many insightful comments and suggestions towards improving
our paper. Below we provide a single response to all the comments of the reviewers, which will be added to the paper.

**FSM-based networks**: FSM-based networks are constructed using weighted linear FSMs (WLFSMs) and perform their computations on
stochastic bit streams. The main computational core of FSM-based networks involves additions and indexing operations. In such networks,
the non-linear combinations of given data are learned by WLFSMs which are more expressive than linear transformations.

**Why are WLFSMs more expressive than linear transformations**: The computing power of linear FSMs comes from a simple fact in
stochastic automata [26]: if elements to a stochastic automaton are from a Bernoulli sequence, the state sequences are then Markov chains
of the first order while the output sequences are Markov chains of generally of higher orders. In stochastic computing (SC), this fact is
interpreted as FSMs whose state transitions are linear, inputs are from a Bernoulli sequence, and outputs are sequences generally of higher
orders.

**Why using binary-stream representations**: The choice of using binary-stream representations in SC-based systems has been made to
accommodate hardware implementations. More precisely, performing computations on bit streams requires simple bit-wise operations,
resulting in a better hardware performance (e.g., power consumption) compared to fixed-point/floating-point computations.

**Unipolar format vs bipolar format**: In FSM-based networks, all the computations are performed on random bit streams where each zero
represents either $-1$ or $0$ depending on the range of the real value that is being represented by stochastic bit streams (i.e., unipolar format
vs bipolar format). For example, the binary sequence of "0 1 0 0" is a stochastic representation of length 4 for both real values of $x = 1/4$
and $y = -2/4$ where $x = (y + 1)/2$. Please note that the conversion (i.e., $x = (y + 1)/2$) allows to follow the flow of information in
floating-point format while the computations in SC are still performed on bit streams of 0 and 1.

**Computations of FSM-based networks**: To visualize the computations of FSM-based networks, we have illustrated the inference
computations of a neuron with two inputs as an example in Figure A where $B(*)$ denotes a Bernoulli function. In this example, the inputs
(i.e., $4/8$ and $6/8$) are represented using unipolar format whereas the output (i.e., $-2/8$) is in bipolar format. Each input individually is
passed to a 4-state to obtain the state values of **s** which are the indices to the weight matrix $W$. Once the weight elements corresponding to
their state values are selected, a Bernoulli sample of them is passed to a scaled adder. The scaled adder performs element-wise additions
and scales the result of additions by a factor of $1/2$ (since there is two inputs). The result of the scaled adder is then sampled to obtain a
binary stochastic stream which could be an input to the next layer.

**Inference latency of FSM-based networks**: In FSM-based network, the length of stochastic streams (i.e., $l$) denotes the latency of the
inference computations.

Figure A: An example illustrating the computations of a two-input neuron in FSM-based networks.

**FSM-based Model for temporal tasks**: Similar to FSM-based networks, FSM-based models for temporal tasks are constructed by
instantiating several WLFSMs in parallel along with two fully-connected layers serving as a transition function and an output generator. In
FSM-based models, WLFSMs are treated as a memory to hold the past information of the data sequence. Unlike FSM-based networks
that perform their computations on stochastic bit streams, the computations of FSM-based models for temporal tasks are performed on
real-valued numbers.

**Why not backpropagating through time**: Since state transitions in FSM-based models occur based on the present state of FSMs and the
present inputs only, the training process (i.e., backpropagation) of such models is decoupled from time. More specifically, FSM-based
models encode the input sequence into a distinct set of state values based on the present entry of the input sequence and the present value of
states. The encoding capacity of FSM-based models is determined by the number of FSMs and their number of states. To better understand
the computational process of such models, let us provide an example for a character-level language modeling task on the sequence of
"b c a" (see Figure B) when considering a dictionary containing the three characters of "a", "b" and "c" only. For the given example, the
state values are initialized with zeros at the beginning. Once the first character (i.e., "a") in a form of one-hot encoding is received, it is
passed to an embedding layer (i.e., a fully-connected layer which is also referred to as transition function) to provide a dense representation
of characters similar to LSTMs. Since WLFSMs only take binary values of 1 and 0 which respectively increment and decrement the state
values by 1, a sample of embedded values is passed to 3-state FSMs. Once the state values are determined, their corresponding weights are
selected and added with the scaling factor (i.e., $\alpha$) of $1/2$. The results of FSM-based layer are then classified using a fully-connected layer
to predict the next character. For the given example, the state value (i.e., **s**) of "0 1" denotes the character "a" whereas the state value (i.e.,
**s**) of "1 2" denotes the sequence of "c a". In fact, the FSM-based model adjusts the weights of the transition function (i.e., $W_x$) such that
FSMs generate a unique set of state values based on the present input and the present state values.

**Why is epoch-to-convergence affected less in FSM-based Models**: Since the training process is decoupled from time and the backprop-
agation is performed based on the current step, the gradients are not compromised by the length of input sequences and consequently
epoch-to-convergence is affected less for long input sequences compared to LSTMs.

**Comparison with other networks**: It is worth mentioning that accuracy performance and hardware performance of GRUs are similar
to those of LSTMs based on our simulations. However, accuracy performance of vanilla RNNs are considerably worse especially when
considering their epoch-to-convergence for long input sequences. We will include the simulation results of GRUs and vanilla RNNs in both
Figure 4 and Table 2 of the paper. Moreover, all the typos raised by reviewers will be fixed and missing literature and explanations will be
added to the final version of the paper.

Figure B: An example of FSM-based model performing a character-level language modeling.

[Meta-Review · NeurIPS 2020]

The paper makes solid contribution with sufficient experiments. Reviewers are on average in favor of accepting the paper.